# OpenReview forum: "ModernVBERT: Towards Smaller Visual Document Retrievers"
_ICLR.cc/2026/Conference — Submitted to ICLR 2026_

### Official Review · Reviewer_SuYD · 2025-10-21

**Soundness:** 4
**Presentation:** 4
**Contribution:** 3
**Rating:** 6
**Confidence:** 3

**Summary:**

This paper systematically investigates the impact of different visual retriever training designs on retrieval performance. The authors analyze the influence of various factors—including model architecture, data scale, attention masking strategies, image resolution, and the scale of image-text pooling in contrastive learning—on different types of visual retrieval tasks. Based on these analyses, they design training methods for the ModernVBERT and ColModernVBERT models. Experimental results demonstrate that the proposed models achieve performance comparable to large-scale models with 10 times more parameters, while requiring only one-seventh of the inference time on CPUs.

**Strengths:**

- A systematic study and analysis of different visual retriever training strategies and their impact on the performance of visual retrieval tasks.
- The proposed ModernVBERT and ColModernVBERT models contain only 250M parameters but achieve excellent performance across multiple tasks, matching the effectiveness of large-scale models with 10 times their parameter count.

**Weaknesses:**

- In Section 3.1, the performance differences between the trained vision-language model and SigLIP across various tasks may be influenced by differences in training data scale and distribution. Further analysis is recommended to strengthen the reliability of the conclusions.
- The experiments only report inference speed on CPUs and do not include results from GPU environments. Considering that models are often deployed on GPUs in practice, the speed differences between models of varying scales may be less pronounced in GPU environments. It is recommended to supplement with relevant tests to provide a more comprehensive evaluation of model efficiency.

**Questions:**

See the Weaknesses.

---

> ### Author Response · Authors · 2025-11-23
> **Rebuttal**
>
> We thank the reviewer for recognizing the quality of our analysis on the factors leading to strong visual document retrieval performance.
>
> **> Strengthening the reliability of experiments in section 3.1**
>
> While the initial experiment aimed to showcase that dual encoders such as SigLIP remained the best solution for natural image representation tasks (even without further training), we followed the reviewer’s suggestions and ran an additional experiment in which SigLIP-base is trained in the same conditions as the VLMs. Results increase across the board, but performance on document retrieval tasks remain significantly inferior to those obtained with VLMs. Furthermore, to study scaling dynamics, we have trained SigLIP large, a 800M parameter model (3.5x more vision parameters than the base models and the VLMs), and show that even then, the smaller VLMs continue to display slightly better results on VDR.  These results have been added to the paper (in red, L261-295 and figure 3).
>
> We further want to point out that the interest in using VLMs instead of dual encoders such as SigLIP, is the fact that it enables many axis of improvements we further exploit in the paper. Typically, we are able to scale the length of the MLM modality alignment phase (L297), swap out the language model backbone to improve capabilities (L407), obtain better control on the image resolution (Table 1), interleave text and image inputs and benefit from cross-modal capability transfer (L384), and crucially, perform multi-vector late interaction search (Figure 5) which requires token-level alignment between text and image models [1]. All of these performance axes do not require larger sets of supervised image-text query data and sum to the large final performance differences observed. We detail this in the modified manuscript (L293).
>
> **> GPU Benchmarks**
>
> A key differentiator of our proposed model is its small size, crucially enabling fast and local deployments without the need for expensive hardware, and maintaining strong performance on cheap hardware, hence the CPU benchmarks. As suggested by the reviewer, we have added GPU latency benchmarks to showcase encoding speeds with higher end hardware and show on H100 GPUs, we are able to embed up to 5000 batched queries per second. Results are mentioned in the main paper (L472) and detailed in Appendix C.6.2.
>
> We thank the reviewer for the thoughtful suggestions, and hope the modifications made to the paper and the extra experiments will satisfy the reviewer’s remaining doubts.
>
>
> ### **References**
> [1] Faysse, Manuel, et al. "Colpali: Efficient document retrieval with vision language models." ICLR 2025

---

### Official Review · Reviewer_hB9o · 2025-10-27

**Soundness:** 2
**Presentation:** 2
**Contribution:** 2
**Rating:** 4
**Confidence:** 3

**Summary:**

The paper studies the problem of visual document retrieval. To this end, the authors have conducted controlled experiments to analyze the factors that can make the performance of visual document retrieval better. This includes attention masking, image resolution, modality alignment data regimes, and late interaction centered contrastive objectives. Based on the conclusions from the controlled experiments, the authors have developed ModernVBERT, a 250M-parameter vision–language model that can achieve good performance on visual document retrieval, though being lightweight.

**Strengths:**

1. Good insights and analysis. The authors have studied several different factors to influence the performance of visual document retrieval models and provided analysis and insights for each factor.

2. An efficient visual document retrieval model is developed. According to the insights and conclusions, the authors have developed a 250M-parameter model which achieves good performance for the task of visual document retrieval.

**Weaknesses:**

1. Application scenarios. It would be better if the authors can discuss more about the potential application scenarios of an efficient visual document retrieval model. This can highlight the importance of developing 'efficiency' for the task. For example, it is important to develop an efficient object detection system as when applied to embodied AI and robotics, it is important to make the model small but effective. Are there wide scenarios where an efficient visual document retrieval model is in a high demand?

2. Conclusions on large models. As the authors have noticed, the paper has only conduced experiments on small models and some of the conclusions made in the paper may not hold for large models. This further limit the scope of the paper, given that the scope of the paper has already been constrained to very specific task - visual document retrieval.

3. Presentation of the paper. It is not good to include fake links in the paper. It is better to say models and code will be made publicly available. In this way the space of the paper can also be saved. The 'finetuning' in the abstract and in the paper should be 'fine-tuning'.

**Questions:**

I would suggest the authors to address my concerns in the weakness section in the rebuttal.

---

> ### Author Response · Authors · 2025-11-23
> **Rebuttal**
>
> We thank the reviewer for their feedback that highlights the strong insights gained through our extensive performance factor analysis, further enabling the release of our lightweight VDR model. We further address the reviewer’s comments:
>
>
> **> Application Scenarios**
>
> Following the reviewer’s comments, we have largely updated the introduction (L33-71) and the abstract (in red) to better contextualize the problem tackled, and to more clearly introduce the task of visual document retrieval (VDR) and its importance.
>
> The updated introduction gives a fuller picture, but in a few words: Visual Document Retrieval (VDR) is a new way to do document retrieval in realistic settings where depending only on extracted text is often not enough—for example with PDFs, charts, tables, and other visually structured content. RAG has been one of the flagship industrial uses of LLMs in recent years, and VDR naturally slots in as a stronger, simpler first-stage retriever. As a result, a wave of VDR models has appeared and gained traction over the last year [1].
> Most of these models are finetuned from large open-source generative vision–language models such as PaliGemma and Qwen2VL, typically with more than 2B parameters. That’s significantly larger than standard text embedding models like BGE-M3 or E5, and it usually forces GPU inference—raising the bar in terms of infrastructure complexity and cost, especially for on-prem or local deployments. This paper bridges that gap: we show, for the first time, that much smaller models can be competitive when trained appropriately, making VDR far more practical to deploy in real-world systems.
>
>
> **> Limited Paper Scope**
>
> As explained above, and in the modified introduction, we believe VDR, although specific, is a task with important and clear practical impact as current models are already widely used today. The findings uncovered also go beyond strictly documents; we report results for natural images, and many of our  findings could be interesting to representation learning in other modalities (impact of bidirectionality and late interaction, signs of positive transfer between modalities, data regimes).
>
> **> Conclusions on Large Models**
>
> Our work deliberately targets the lower end of the model size spectrum, where the benefits for local and resource-constrained deployments are most immediate. Within our budget, we were not able to train larger models using the same recipe, and we leave an exploration of that regime to future work. That said, we see no fundamental reason why our main findings—on the role of bidirectionality, cross-modal performance transfer, and the impact of input resolution—would not carry over to larger model scales.
>
> **> Presentation of the paper.**
>
> Thank you for the suggestions. We have modified the spelling and temporarily removed the link. We clarify that models and codes are already publicly available in the public version of this work.
>
>
> We thank the reviewer once more for their insightful feedback and believe modifications made have significantly improved the paper.
>
> ### References
> [1] Faysse, Manuel, et al. "Colpali: Efficient document retrieval with vision language models." ICLR 2025

---

### Official Review · Reviewer_JWzt · 2025-10-31

**Soundness:** 2
**Presentation:** 3
**Contribution:** 2
**Rating:** 4
**Confidence:** 4

**Summary:**

The paper propsoes ModernVBERT, small but powerful mutimoal embedding model. Unlike larger multimodal embedding embedding models, ModernVBERT opts for a text encoder backbone instead of a decoder LLM backbone. The paper demonstrates ModernVBERT's profinciency in visual documents retrieval on VideoRe, matching the performence of substancially larger models.

**Strengths:**

- Using encoder backbones in multimodal embedding models is under-explored, and the results showing encoder backbones are better then decoder backbones are interesting.

- Small document retrievers, especially those that can run on CPUs, are an interesting and distinct setting. Existing work focuses almost exclusively of the >7B range, with is not feasible for most CPUs.

- The writing is mostly clear, albeit with a few typos.

**Weaknesses:**

- Most of the experiments and conclusions in section 3.1 are are severely confounded by the MLM being trained on a mixture containing a very large proportion of documents compared to SigLIP. This results in potentially misleading conclusions.

- Although the model has less parameters than other small encoders, the model’s resolution scaling scheme significantly increases the model’s compute requirement by tokenizing an image into more visual tokens (although only on the embedding side).

- The tradeoff between ColModernVBERT and a larger CLIP model that has been specialized for document embedding is unclear. In general, the baselines considered are either too large to be a fair comparison or not specialized for document retrieval.

- BiModernVBERT is very similar in approach to existing mulitmodal embedding models, with the main difference being the use of an encoder instead of a decoder LM.

- The manuscript does not contain an explanation for how the resolution of images is “scaled” in the image tokenizer.

- Although ModernVBERT has strong document embedding capabilities (for its size) the experiments are not sufficient to conclude what aspect of its design this results from. It could be the training data, architecture, or even the batch curation strategy acting as implicit hard negative mining.

Minor:
- missing space: “(LoRA)(Hu et al., 2021)”

- typo: “alignement”

**Questions:**

- How is the resolution scaled in the vision backbone and how does that effect the number of visual tokens?

- The post training phase mentions that it uses 118M(illion?) samples from MSCOCO, which I assume is actually 118k (judging by table 7)?

- Can you control for data when compared to a dual encoder (I.e. SigLip) baseline?

- What is exactly is meant by "too large to run on CPUs" (table 3)? Although the models may be slow to run on CPU, they should still run.

---

> ### Author Response · Authors · 2025-11-23
> **Rebuttal: Part 1**
>
> We thank the reviewer for his suggestions and careful reading of our work that underlines its focus on an underexplored topic and its interest. We further address weaknesses and questions point by point:
>
> **>Strengthening the reliability of experiments in section 3.1**
>
> While the initial experiment aimed to showcase that dual encoders such as SigLIP remained the best solution for natural image representation tasks (even without further training), we followed the reviewer’s suggestions and ran an additional experiment in which SigLIP-base is trained in the same conditions as the VLMs. Results increase across the board, but performance on document retrieval tasks remain significantly inferior to those obtained with VLMs. Furthermore, to study scaling dynamics, we have trained SigLIP large, a 800M parameter model (3.5x more vision parameters than the base models and the VLMs), and show that even then, the smaller VLMs continue to display slightly better results on VDR.  These results have been added to the paper (in red, L261-295 and figure 3).
>
> We further want to point out that the interest in using VLMs instead of dual encoders such as SigLIP, is the fact that it enables many axis of improvements we further exploit in the paper. Typically, we are able to scale the length of the MLM modality alignment phase (L297), swap out the language model backbone to improve capabilities (L407), obtain better control on the image resolution (Table 1), interleave text and image inputs and benefit from cross-modal capability transfer (L384), and crucially, perform multi-vector late interaction search (Figure 5) which requires token-level alignment between text and image models [1]. All of these performance axes do not require larger sets of supervised image-text query data and sum to the large final performance differences observed. We detail this in the modified manuscript (L293).
>
> **>Clarifications on Image Resolution and Image scaling**
>
> As described in L162, our approach follows recent work where images are downscaled (or upscaled) so that the lengths and widths reach a multiple of 512 pixels to preserve the aspect ratio, with padding being used on the smaller side when necessary. It is then possible to feed the 512 x 512 pixel patches to the SigLIP encoder in parallel, which produces 1024 patch embeddings that is reduced to 64 through pixel shuffling. As explained in L153, we also preserve a global view of the image by downscaling and padding it to 512 x 512 pixels and feeding through the SigLIP.
>
> To illustrate, a 1024 x 1000 image would be padded to 1024 x 1024, then split into four 512 x 512 patches, producing 64 x 4 = 256 image tokens. To this, we add the 64 tokens obtained from  the global image, for a total of 320.
>
> We are thus able to play on the image resolution by scaling images to different sizes. A 2048x2048 image results in (2048/512)**2=16 patches, so 16*64 + 64 = 1024 +64 tokens which is in line with the token amounts in ColPali. We show in Table 1 that resolution is a compute/performance tradeoff. While the final resolution of ModernVBert is 2048 pixels, running it with 1024 pixels only leads to a loss of 3.6 points on document retrieval tasks. We have added clearer explanations to the manuscript and further details in Appendix C.6.1.
>
>
> **>Comparisons ColModernVBERT and a larger CLIP**
>
> As detailed in this rebuttal (*>Strengthening the reliability of experiments in section 3.1*), we have followed the reviewer’s suggestion and fine-tuned a larger SigLIP2 model (881M parameters, 3.5x more vision parameters than ModernVBert). The trained bigger dual encoder displays performances slightly below those of the smaller encoder VLM trained in similar conditions. As mentioned above and importantly, most of the axes of performance we have found crucial to further improve our model are not applicable (improving backbone LLM, longer modality alignment, multi-vector late interaction, image resolution, cross-modal positive transfer). These gains are obtained independently of the supervised image:query contrastive training dataset (the data bottleneck), and unlock dozens of points of performance. We indicate this L293 of the revised manuscript.
>
> **>Inclusion of baselines that are too large or not specialized for document retrieval**
>
> In the paper, we report results for ColFlor, which is to our knowledge, the only other effort to train Visual Document Retrieval models with <1B models. ColModernVBert largely outperforms it, and performs well enough to perform comparably or better than much bigger models, typically ColPali or VLM2Vec (10x the size) published at ICLR 2025.
> We have reported the most relevant baselines we could think of, and extended the list to include non-specialized smaller models, and much larger models. We have modified Table 3 to clarify that JinaClip, BGE-M3-Visualized and SigLIP2 while partially trained on documents, are not specifically trained for VDR.

---

> ### Author Response · Authors · 2025-11-23
> **Rebuttal: Part 2**
>
> **> BiModernVBERT is very similar in approach to existing multimodal embedding model**
>
> BiModernVBert results are indicated for comparison. While not our best performing model, it sets a strong single-vector retrieval baseline for its size, and we believed it would be interesting to report single vector results at this model size, but this is not the focus of our paper, nor is it mentioned as a contribution.
>
> **> Explanation for how the resolution of images is “scaled” in the image tokenizer.**
>
> Explanations of how images are tokenized are given L152-156 and are detailed above in *>Clarifications on Image Resolution and Image scaling.*
> Images are not scaled in the image tokenizer, they are downscaled (or upscaled) in pixel space using standard image processing libraries (Pillow), and then processed by the image tokenizer. We have added details in the manuscript (in red, L360).
>
> **> Identifying what aspects of the design performance stems from**
>
> Section 3 aims at comparing in rigorous conditions where performance stems from. In all ablations, we change a single criteria and observe results on downstream performance.
> Notably, we observe (1) the impact of the attention mask (2) the impact of the modality alignment scale (3) the impact of image resolution (4) the impact of adding cross-modal data in the contrastive training (5) the impact of late interaction and multivector embeddings. In appendix C, we further ablate (6) the impact of contrastive training duration (7) various contrastive training data mixes. We believe this paints a detailed picture of where performance stems from.
> Furthermore, as detailed in >Strengthening the reliability of experiments in section 3.1, we have run additional experiments to yield a more complete picture of the Figure 3 ablation with trained SigLIP baselines.
>
> ##  Reviewer Questions
>
> > How is the resolution scaled in the vision backbone and how does that effect the number of visual tokens?
>
> See *>Clarifications on Image Resolution and Image scaling*
>
> > The post training phase mentions that it uses 118M(illion?) samples from MSCOCO, which I assume is actually 118k (judging by table 7)?
>
> This is correct, thank you very much for catching this typo. We have made the modification, along with the other typos reported.
>
> > Can you control for data when compared to a dual encoder (I.e. SigLip) baseline?
>
> Thank you for the suggestion. We have done so as detailed in  *>Strengthening the reliability of experiments in section 3.1.*
>
>
> > What is exactly meant by “too large to run on CPUs” (table 3)? Although the models maybe slow to run on CPU, they should still run.
>
> Our intention was to indicate that running these models on CPU were impractical. Our criterion for this was whether a query would be embedded in less than 10 seconds on a Google Colab CPU (12Gb RAM, fairly standard for low-end CPU servers). As the larger models would not fit in RAM, they required memory offloading, leading to high latency (in the order of several seconds). Following the reviewer’s comments, we have rerun the benchmark on high RAM setups in order to properly run all models without bottlenecks. To further improve our work, we have also added GPU latency benchmarks as well (in red, L470-475 and appendix C.6.2).
>
>
>
> We thank the reviewer again for their comprehensive feedback, and hope their concerns were addressed in our explanations and further experiments.
>
> ### References
> [1] Faysse, Manuel, et al. "Colpali: Efficient document retrieval with vision language models." ICLR 2025

---

### Official Review · Reviewer_SfZz · 2025-11-13

**Soundness:** 4
**Presentation:** 2
**Contribution:** 3
**Rating:** 6
**Confidence:** 3

**Summary:**

It took me a long while to write this review.

Because I found the argument of this paper so boring and far from my interests, and I have been given really too many papers to review. Still, I got assigned this paper and here I am.

Now let me be clear: by saying that I found the argument boring, I do not mean that it is bullshit. It actually tackles a very concrete problem and offers a sensible improvement.

A problem so important that I myself, without even noticing it, make constant use of the currently available solutions.

Recently I saw a cool video about the conception of the IKEA Lack table, the small coffee table you can buy for 10$. It is a cheap coffee table, nothing exactly exciting or groundbreaking, but it is something fundamentally useful, and the idea of making it out of hexagonal cardboard had quite an impact by making it astonishingly cheap and allowing its wide spreading all over the world.

Tables may be boring, but this accomplishment deserves attention, and the details of the idea are worth knowing and celebrating. I see this paper as the blueprint of the IKEA Lack table, containing the very clever innovations and their reason, such as the hexagonal cardboard internal frame (the patch representations) or the trick of not using full-wood legs (the bidirectional attention à la BERT).

My criticism is about the presentation.

The paper should do a better job explaining the problem and showing why it is relevant. ICLR is full of people who do not really care about tables, and since your paper is here, you should make an effort in explaining the relevance of what you have done, perhaps by doing a better job explaining the problem in the abstract and the introduction.

Already the abstract was cryptic for me. Do not rely on the tag “Visual document retrievers”; most people don’t know what you are talking about. Your paper should be more self-contained.

I have nothing to say regarding the rest; I find your paper well executed.

I will put a 6 that I will update to an 8. I hope to update it after having read a new abstract and introduction that make me more passionate and aware of the small but very significant improvement that is going to be described in the next pages.

This is the video of the IKEA Lack table: https://www.youtube.com/watch?v=0h8vAGCiRX0

**Strengths:**

see above

**Weaknesses:**

see above

**Questions:**

see above

---

> ### Author Response · Authors · 2025-11-23
> **Rebuttal**
>
> We thank the reviewer for his time, suggestions and fun metaphor. To sum up, the reviewer finds the paper solid, offering concrete solutions to a widespread problem. The reviewer however suggests the introduction lacks contextualization for readers outside the field of document retrieval to better apprehend the problem.
>
> We have thus very largely modified the introduction  (L33-71) and the abstract (in red) to better motivate our work and make it more accessible to readers outside of the field of document retrieval. To reduce rebuttal redundancy, we further link [the rebuttal to reviewer’s hB9o](https://openreview.net/forum?id=zifQaLogHV&noteId=1qwS8EFMFN) (*> Application Scenarios*) for an overview of why Visual Document Retrieval is a novel yet exciting task with much practical importance.
>
> We hope our edits will satisfy the reviewer.
>
> *PS: Amazingly, the laptop used to type this review is currently sitting on the exact IKEA table used in the metaphor.*

---

### Author Response · Authors · 2025-12-02
**Summary of Review Period for the AC**

We briefly summarize the reviews and the discussion period to facilitate the AC’s assessment of our work.

**Strengths:** The reviewers describe the paper as a *well-executed*, *systematic* and *insightful* study (SfZz, hB9o, SuYD) which identifies performance factors in visual retrieval models, notably through *interesting* and *under-explored* directions (JWzt) and *clever innovations* (SfZz). The models openly released in this work, which are small enough to run on CPU, are deemed *distinct and interesting* (JWzt),  achieving *excellent* performances by matching much bigger models (SuYD, hB9o) and *offering a sensible improvement to a very concrete problem* (SfZz).

**Reviewer Comments:** We have clearly addressed all reviewer comments in the individual rebuttals and updated the paper (in red) to reflect the suggested improvements. Concretely:

| Suggestions | Modification made | Outcome |
|--------------|-------------------------------|----------|
| **Additional Ablation Baseline:** Reviewers (JWzt, SuYD) were interested in section 3.1's dual encoder baseline trained on the same data.| **New Experiment:** We trained the requested model (+ a much bigger variant) under identical conditions.| Results *strengthen* our claim that small VLMs outperform much larger dual encoders on documental tasks. (Sec 3.1)|
| **Broaden audience:** Reviewers (SfZz, hB9o) suggest a simpler introduction and contextualization of the “Visual Document Retrieval” task for a more general audience. | **Revision:** We have used the extra page to update the Abstract & Introduction sections (L33–71).| The task relevance and application scenarios of our work are more clearly motivated. SfZz originally stated they would *improve the rating to 8* once we did so.   |
| **Latency Benchmarks:** Reviewers (JWzt, SuYD) suggested clarifications and additional GPU latency benchmarks.  | **Extended Experiments:** We added GPU latency benchmarks, and detailed CPU latency measures. (Sec 4.2, Table 12 in C.6.2) |  New results further *strenghten* our efficiency claims by demonstrating the performance improvements w.r.t. baselines *increase* on GPU.|
| **Request for Technical Clarifications:** Reviewer JWzt had questions on resolution scaling and our choice of baselines | **Further Explanations:** While information was present in the original manuscript, we added further details and given concrete examples in the updated version. (Sec 3.2, 4.2,  Appendix C.6.1) |  These details (with the latency benchmarks) further clarify that our model does not produce more visual tokens than comparable baselines |

We are confident all reviewer comments have been clearly addressed and believe suggestions made in the review process have strengthened our work. We thank all reviewers and ACs for their time and relevant feedback.

---

### Meta-Review · Area_Chair_7dmi · 2026-01-05

**Summary:**

The paper introduces ModernVBERT, a 250M-parameter vision-language encoder designed for efficient Visual Document Retrieval. The main contribution lies in utilizing a smaller text encoder backbone (instead of repurposing a large decoder backbone) to achieve high performance while remaining efficient enough for CPU-based inference. The authors provide a systematic study of the training pipeline, analyzing how factors such as attention masking, image resolution, and modality alignment impact retrieval success. An extensive comparative analysis is provided that demonstrates the effectiveness of ModernVBERT with respect to other small and large models.

The main factors contributing the the final recommendation are:

+ **Presentation problems**: Multiple reviewers commented on problems with presentation in the submitted manuscript. These range from typos to more significant, overarching problems with the presentation of the motivations and technical content of the paper.
+ **Possibly unfair comparative analysis**: Again, multiple reviewers commented on the methodology employed for the comparative performance analysis. They note that the training data for ModernVBERT is not aligned with that used for some baseline models (e.g. SigLIP). This points to a systematic problem, which it that to effectively justify the empirical findings of the paper it is necessary to train alternative architecture under identical training conditions.

While the reviewers recognize that the results presented in the submitted manuscript are interesting, the paper is essentially an empirical exploration of the training and architectural hyperparameters of Visual Document Retrievers. While certainly useful, the technical contribution of the work is limited, and the theoretical contribution non existent. As such, the work does not represent broad interest to the ICLR community and the recommendation is to Reject.

**Reviewer Concerns:**

The main concerns expressed by the reviewers include:

+ **Presentation and Motivation:** Multiple reviewers commented on problems with presentation and the primary motivations behind the work. These concerns were only partially addressed in rebuttal, and a more thorough revision of the work would be necessary to completely address them,
+ **Confounding Experimental Factors:** reviewers JWzt and SuYD expressed concern that the performance differences between the proposed vision-language model and baselines like SigLIP might be confounded by differences in training data scale and distribution rather than architectural improvements. This was only partially addressed in rebuttal, but a much more thorough exploration of multiple model architectures trained under identical conditions is needed.
+ **Efficiency and Compute Trade-offs:** reviewer JWzt pointed out that while the model has fewer parameters, its resolution scaling scheme significantly increases compute requirements by generating a high number of visual tokens. Questions regarding efficiency (including GPU inference timings) were adequately addressed in reubuttal.

**Reviewer Scores:**

+ **R1 (SfZz)**: A vacuous and irresponsible review that contributes nothing to the discussion. While empirical evaluations and explorations have their place, the ICLR main conference is not that place.
+ **R2 (JWzt)**: Primarily concerned with presentation problems, fairness of comparative performance analysis, and ablations. This reviewer would probably have been convinced by rebuttal clarifications regarding ablations, but I doubt they would have been convinced to significantly change their score based on the rebuttal regarding fairness of comparisons.
+ **R3 (hB9o)**: Again, most concerns related to presentation and and broader conclusions possible to draw from the comparative analysis. This reviewer would probably have been satisfied with the rebuttal related to application scenarios, but it is unclear if they would have significantly increased their score.
+ ** R4 (SuYD)**: Once again, questions regarding the fairness of comparison with SigLIP due to the significant differences in training data distribution. I do not believe this reviewer would have been convinced to significantly change their score.

In summary: There is no serious, enthusiastic support for accepting this paper on the part of the reviewers.

---

### Decision · Program_Chairs · 2026-01-26

Reject